# Challenges in Using Circulating Micro-RNAs as Biomarkers for Cardiovascular Diseases

**DOI:** 10.3390/ijms21020561

**Published:** 2020-01-15

**Authors:** Kyriacos Felekkis, Christos Papaneophytou

**Affiliations:** Department of Life and Health Sciences, University of Nicosia, 2417 Nicosia, Cyprus; felekkis.k@unic.ac.cy

**Keywords:** circulating micro-RNAs, cardiovascular disease, biomarkers, prognosis

## Abstract

Micro-RNAs (miRNAs) play a pivotal role in the development and physiology of the cardiovascular system while they have been associated with multiple cardiovascular diseases (CVDs). Several cardiac miRNAs are detectable in circulation (circulating miRNAs; c-miRNAs) and are emerging as diagnostic and therapeutic biomarkers for CVDs. c-miRNAs exhibit numerous essential characteristics of biomarkers while they are extremely stable in circulation, their expression is tissue-/disease-specific, and they can be easily detected using sequence-specific amplification methods. These features of c-miRNAs are helpful in the development of non-invasive assays to monitor the progress of CVDs. Despite significant progress in the detection of c-miRNAs in serum and plasma, there are many contradictory publications on the alterations of cardiac c-miRNAs concentration in circulation. The aim of this review is to examine the pre-analytical and analytical factors affecting the quantification of c-miRNAs and provide general guidelines to increase the accuracy of the diagnostic tests in order to improve future research on cardiac c-miRNAs.

## 1. Introduction

Cardiovascular diseases (CVDs) are one of the leading causes of death in developed countries, highlighting the need to identify novel prognostic and diagnostic biomarkers for the prevention and treatment of such diseases [1,2]. However, the identification of reliable biomarkers that could routinely be measured in serum or plasma remains a major challenge in CVDs [3]. Recently the attention of the scientific community has been turned to the identification and validation of circulating micro-RNAs (c-miRNAs) as diagnostic and therapeutic biomarkers for CVDs because several miRNAs appear to be essential regulators of cardiovascular system physiology and development [4]. Specifically, in the cardiovascular system, miRNAs regulate basic functions in almost all cell types including cardiac muscle, endothelial cells, smooth muscle, inflammatory cells, and fibroblasts and, therefore, play a pivotal role in the pathogenesis of several CVDs. In general, miRNAs are single-stranded small (18–24 nt) non-coding RNA molecules involved in gene expression regulation in numerous biological processes and disease and therefore called “master regulators” of gene expression [5]. miRNAs are secreted by cells in (i) a free form, i.e., not bound to any biomolecule [6]; (ii) a complex with RNA-binding proteins including Argonaute 2 protein-AGO2 [7], high-density lipoprotein-HDL, or nucleophosmin I-NPM1 [6]; (iii) cell delivered vesicles such as microvesicles [8] and exosomes [9]. The difference between protein-associated and vesicle-enclosed remains elusive. It has been proposed that the secretion mechanism of each miRNA depends on the cell type from which it is released as well its fate and function [10]. For example, miRNAs detected only in the HDL-associated fractions may be generated by cells with lipoprotein transport pathways. Interestingly, it has recently been demonstrated that inflammatory cells including macrophages, monocytes, and neutrophils export miRNAs to HDL, as many of the miRNAs that are highly abundant in these types of cells have been detected in elevated concentrations in HDL [11]. miRNAs associated with other types of RNA-binding proteins, including AGO2 and NPM1, may also be actively released from donor cells and taken up by recipient cells. On the contrary, miRNAs that are transported with vesicles may originate from specific cell types that produce large amounts of vesicles [12]. The biogenesis, function, and circulation of miRNAs have been extensively reviewed elsewhere [13,14,15].

Cell-secreted miRNAs facilitate the exchange of genetic information between cells and play an important role in cell-to-cell communication. They are also implicated in physiological processes such as the regulation of immunity and angiogenesis or cellular migration, while they are also involved in various pathological conditions [16]. Importantly, miRNA released from cells can be detected in a plethora of human body fluids including saliva, urine, blood, serum, plasma, seminal fluid, urine, and pleural effusion [17].

Even though several biological molecules, including cytokines, peptides, proteins, and metabolites are currently being evaluated as potential prognostic and diagnostic biomarkers for CVDs [18], c-miRNAs possess many attractive features of biomarkers especially due to their stability in the blood circulation since they are not degraded by endogenous RNases [19,20]. In general, an ideal biomarker must: (i) be accessible using non-invasive methods; (ii) exhibit a high sensitivity and specificity to the disease; (iii) be detected early; (iv) exhibit sensitivity to changes in the disease; (v) be stable, i.e., a long half-life within the sample is required; and (vi) be accurately and rapidly detected [21]. In addition, the expression of miRNAs is often tissue- or disease-specific [22], while they can be easily detected in plasma/serum samples using sequence-specific amplification techniques and thus c-miRNA are suitable for non-invasive analysis in CVD-patient samples [23]. Moreover, miRNAs, are extremely stable in plasma/serum and resistant to some hard conditions (discussed further below) including low and high pH, boiling, and long storage and can withstand repetitive freezing and thawing cycles [24,25].

Several studies have highlighted the potential of c-miRNAs as biomarkers for the early diagnosis of CVDs while others have demonstrated their potential as tools for prognosis and treatment interventions of several CVDs [26]. In pathological conditions, including CVDs or cancer, there are specific patterns of c-miRNAs that can be employed for monitoring and/or diagnostic purposes. Since different features of a disease might result in altered profile and/or concentrations of plasma/serum miRNAs, a combination of miRNAs will provide a more sensitive and specific diagnostic. Moreover, they could provide clues for the intermediate endpoints in clinical trials [27]. Despite the substantial progress in identifying the association of specific c-miRNAs in various diseases including CVDs, factors that may affect the measurement of their concentrations have yet to be fully addressed [28]. Additionally, due to their low concentration in plasma and serum, miRNAs can only be quantified using amplification techniques and thus the amount of starting material and the extraction method may affect the results [28]. It has been demonstrated that sample type and processing, as well as RNA extraction methods significantly affect the miRNA profile and/or expression levels [28,29]. Importantly, a comparison of c-miRNA profiles obtained by different laboratories for the same disease, including CVDs, revealed different expression levels of specific miRNAs (see [30] and references cited therein), illustrating the need to standardize the detection and analysis of c-miRNAs. Another major issue regarding the discovery and validation of cardiac c-miRNAs as biomarkers for CVDs is that most studies performed to date used small cohorts (<100 subjects). Despite the fact that several miRNAs were confirmed as good biomarker candidates in more than one study, further validation in larger patient cohorts is needed [31].

In this review, the pre-analytical and analytical challenges during the determination of c-miRNAs concentrations will be discussed. Initially, we will summarize the current knowledge regarding circulating miRNAs as potential biomarkers in CVDs. Subsequently, we will discuss the effect of pre-analytical and analytical variables on c-miRNAs expression levels and address some of the challenges and opportunities.

## 2. Circulating miRNAs as Biomarkers in Cardiovascular Diseases 

CVDs include several diseases such as coronary artery disease (CAD), hypertension, hyperlipidemia, congestive heart failure (CHF), stroke, cardiac hypertrophy (CH) and arrhythmia [32]. Diagnosis and risk stratification of patients with heart failure remains a challenge [33]. The identification of specific miRNAs in the cardiovascular system has opened a new field to elucidate the molecular mechanisms controlling the expression of genes in a healthy cardiovascular system. MiRNAs play a pivotal role in a wide range of biological processes, including several features of normal cardiovascular development and physiology [34,35]. Importantly, cardiac miRNAs are potential therapeutic targets in cardiac and vascular disease, while they could be used as novel diagnostic biomarkers for CVDs [34,36]. More than 2000 miRNAs have been identified so far; however, the biological functions of many of them are still not adequately investigated. MiRNAs influence the production of proteins, i.e., they silence the expression of genes by regulating the translation of messenger RNAs by base-pairing with the 3′ UTRs. MiRNAs participate in normal cellular physiology such as differentiation and proliferation but also in the progression of a variety of diseases by influencing gene expression in heart, lung, immune system, etc. [37]. Both miRNA biosynthesis and function have been extensively reviewed [38,39]. The role of c-miRNAs in the cardiovascular system as well as their function, and potential diagnostic/therapeutic uses have also been previously reviewed [40] and therefore will not be discussed here.

MiRNAs (e.g., miR-1, miR-21, miR-29a) have been identified in cardiac tissue at all stages of development [41,42] and probably play an essential role in both normal cardiac maintenance and disease [42,43,44,45]. In addition, a large proportion of miRNAs are expressed as clusters that are subsequently processed into individual mature molecules. It has been demonstrated that specific miRNA clusters are essential to the normal development of the cardiovascular system [37]. For example, some miRNAs clusters are important for activation of endothelial cells, post-natal neovascularization, cardiovascular development and regeneration [37,46] as well as for vascular smooth muscle cell functions such as migration, plasticity, and contractile ability [47]. Other miRNAs have been implicated in the regulation of cardiac contractility, fibrosis [48], and the balance between the α- and β-myosin heavy chains (MHC) [49]. Dysregulation of miRNA clusters leading to altered biological functions is key to the pathogenesis of several CVDs [50,51]. Since the discovery that miRNAs are found in the circulation, they have been investigated as novel biomarkers for CVDs [52]. c-miRNAs have been proposed as biomarkers for coronary artery disease (CAD) [53], acute myocardial infarction (AMI) [54], hypertension [55], heart failure (HF) [56,57], viral myocarditis (VM) [58], atherosclerosis, hypertrophy [37], acute coronary syndrome (ACS), ST-elevation myocardial infarction (STEMI) [59], electrophysiological abnormalities [60,61], and hypertrophic cardiomyopathy [62]. Nevertheless, a more comprehensive understanding of c-miRNA will lay the foundation for the development of c-miRNA-based diagnostic and therapeutic interventions for cardiovascular diseases [63].

However, to date, there is no accurate, consistent and robust method for detecting and measuring c-miRNAs in serum/plasma samples as summarized in Table 1, highlighting the importance of having a general guideline as to how samples are prepared and handled for determination of circulating miRNA levels. 

The most important pre-analytical, analytical, and other specific technical issues that should be addressed during the evaluation of miRNAs expression levels in blood samples including plasma and serum are discussed in the following paragraphs. 

## 3. Challenges in Using Circulating miRNAs as Biomarkers in Cardiovascular Diseases

As we discussed above, c-miRNAs could make excellent (i) biomarkers for prediction and prognosis, and (ii) therapeutic targets for CVDs, and despite their numerous advantages as biomarkers, there are still several issues to overcome before their clinical application. The most essential technical issues are the isolation and purification of samples since the integrity and purity of RNA are the basis of detection and quantification. Analysis of c-miRNAs for CVDs requires the careful consideration of the unique properties of body fluids that can make the reproducible and quantitative assessment of RNA challenge. For example, enzymes involved in the amplification and analysis of RNA can be affected by blood components that co-purify with miRNA. Thus, if miRNAs are to be effectively utilized as biomarkers, it is important to establish standardized protocols for blood collection and miRNA analysis to ensure accurate quantitation [28].

A major challenge in c-miRNAs biomarker discovery and validation for several diseases, including CVDs, is the biological complexity of the mechanism of the disease pathogenesis. There are also many technical issues hindering this process, such as the lack of standardization in sampling, processing, and storage of samples. Variations in the amount of starting material and isolation method to obtain miRNA may introduce bias and contribute to quantification errors [72]. 

### 3.1. Effect of Sample Selection and Processing in Evaluating the Expression Levels of c-miRNAs

The evaluation of miRNAs in circulation begins with the collection of blood samples from CVD- patients and/or from healthy individuals. As mentioned above, the selection of blood fraction (whole blood, plasma, or serum) and handling/storage conditions have a great impact on the miRNAs profile and several factors during this early stage affect the quality of the results that are usually overlooked. The effects of both the selection of blood fraction and anticoagulant on the quality of the results are highlighted in the example of Table 2. 

In the example given in Table 2, three research groups compared the expression levels of miR-1 between AMI patients and healthy individuals. miR-1 is highly expressed in the heart and plays a protective role against HF and cardiac hypertrophy by regulating several hypertrophy-associated genes including transcription factors, receptor ligands, apoptosis regulators, and ion channels [40,67,76,77]. However, the expression levels of miR-1 in AMI patients, (expressed as increase-fold compared to healthy individuals) varied in these studies (Table 2). The discrepancy among the results can be partly explained by the fact that each group used a different blood fraction (plasma or serum) while the two groups that determined the miR-1 in plasma samples used a different type of anticoagulant. Specifically, Liu et al. [73] used collection tubes containing EDTA while in the study of Zhong et al. [74] tubes containing citrate were used from plasma isolation. In the third study, Li at el. [75] evaluated the expression levels of miR-1 in serum samples from AMI patients and healthy individuals. In EDTA-treated plasma, a 2.8-fold increase of miR-1 expression levels was recorded while in serum samples only a 1.5-fold increase of miR-1 concentration was determined. Surprisingly, in citrate-plasma, a 60-fold increase of miR-1 was recorded that is ~21 and 40 times higher compared to the EDTA-plasma and serum, respectively. It should be noted that in each study, a different extraction and detection method was employed (discussed further below), while the cohorts that were used differ in terms of the number of AMI patients and healthy individuals.

Overall, based on the above example, it could be suggested that results from studies using different blood fractions and blood tubes should not be directly compared to each other and most importantly only miRNAs that are not marginally up- or down-regulated will be suitable as clinical biomarkers.

#### 3.1.1. Sample Selection: Plasma or Serum?

The sample source is one of the most critical aspects of c-miRNA levels determination [78]. The expression of miRNAs is different between the samples extracted from the serum and plasma even from the same individual. In addition, whole blood should be avoided because cellular fraction will also contribute miRNAs (Figure 1) [79].

Plasma is usually preferred over serum in studying c-miRNAs because, during the coagulation process, RNA molecules are released and may change the true profile of circulating miRNAs. However, plasma contains cellular components that may contribute miRNAs from apoptotic or lysed cells (e.g., red blood cells-RBCs, platelets), as well as anticoagulants such as heparin that can inhibit downstream methodologies (discussed further below), and therefore serum is considered the best fraction for c-miRNA detection [80,81].

However, there is an inconsistency in the results published by different groups when comparing the levels of the same miRNAs (i.e., miR-15b, miR-16, and miR-24) in plasma and serum from healthy volunteers (Table 3). Wang et al. [78] reported that the total miRNA concentration was higher in serum than in plasma, probably due to the release of miRNAs from platelets and blood cells. Moreover, the difference between concentrations of miRNA in serum and plasma indicated a correlation with miRNAs from platelets, suggesting that the coagulation process may affect the profile of extracellular miRNA in blood [78]. However, in another study, McDonald et al. [82] reported that the total miRNAs concentration was higher in plasma than in serum, while Mitchell et al. [83] described the concentration of the test miRNAs in plasma and serum to be similar. Overall, it would be interesting to examine whether this discrepancy is related to hemolysis or other technical artifacts. It has been previously demonstrated that handling bias can occur during the extraction step [82] and plasma miRNAs may be contaminated by cellular miRNAs, as well as hemolysis [82,84,85]. Importantly, the intrinsic variability of c-miRNAs levels is often overlooked [82] and may affect the quality of the results.

It has been proposed that the c-miRNA concentrations differ in plasma and serum due to the RNA/miRNA “trafficking” between cellular compartments and the extracellular environment (see [78] and references cited therein). During the coagulation process, blood cells are exposed to “stressful conditions”, which may “promote” the release of specific miRNAs/ RNAs, as was observed when cells were exposed in vitro to serum-free conditions [6]. In addition, during the coagulation process, significant amounts of miRNA that are found in platelets [86] might also be released into the serum. Interestingly, the coagulation process also affects sample-to-sample variations on the protein profile, which makes data analysis and comparison between different studies difficult [87]. Even though it has previously been suggested by Wang et al. [78] that plasma should be used to evaluate the levels of c-miRNA, because RNA released during the coagulation process may alter the true profile of miRNA in circulation, the majority of the archived samples are stored as serum, and therefore, special attention must be given to this difference. 

#### 3.1.2. Effect of Anticoagulant 

The choice of anticoagulant (EDTA, heparin, citrate) that is used for plasma isolation may also have an impact on miRNAs expression levels (Figure 1). For example, it is well known that heparin inhibits the reverse transcriptase and polymerase enzymes used in qRT-PCR [88]. In the case that the use of heparin is required, samples must be treated with heparinase prior to analysis to increase miRNA detection [89]. Similar to heparin, sodium citrate might also affect PCR results [90,91], and therefore blood collection tubes containing EDTA are recommended over sodium citrated and heparin for PCR-based assays [92]. It has also been reported that the use of EDTA improves the quality of miRNA expression profiling compared to sodium citrate [53]. Interestingly, several studies have shown that miRNA concentrations derived from citrated-treated plasma were lower compared to those obtained from EDTA-treated plasma from the same individuals [53,91]. However, in another study, it was suggested that citrate probably increases the sensitivity of miRNA detection compared to EDTA [81]. Furthermore, it has been reported that the use of EDTA as an anticoagulant inhibits RNA precipitation leading to lower yields, though this effect has not been reported with silica columns that are widely used in commercially available miRNA isolation kits [93].

In conclusion, expression levels of miRNAs that have been obtained from blood samples collected from different blood fractions or in the case of plasma from tubes containing different anticoagulants should not be included in the same study or directly compared to each other, to avoid conflicting results [92].

#### 3.1.3. Effect of Centrifugation

Another important factor that is usually overlooked and may have an impact on the miRNA profile in both the serum and plasma samples is the centrifugation conditions such as the applied force, centrifugation time, temperature acceleration, etc. [94]. In addition to sample collection and transportation, serum or plasma recovery by centrifugation is one of the most crucial steps in the pre-analytical phase. Isolation of plasma and serum samples is generally carried out using different centrifugation conditions. However, the centrifugation force, type or rotor, time and speed are important variables. It has been demonstrated that the centrifugation force during plasma recovery affects the levels of c-miRNAs due to the number of platelets remaining in the sample [95]. Previous studies have demonstrated that alteration of centrifugation conditions could result in the recovery of platelet-poor or platelet-rich plasma/serum, which could lead to contamination of plasma with different amounts of blood cells [96]. In detail, prolonged centrifugation at a high speed may cause hemolysis and therefore release of miRNAs from platelets while low speed and brief centrifugations may lead to poor separation of serum or plasma from cellular components [97].

In the example of Table 3, each group used a different centrifugation protocol to isolate plasma or serum. Mitchel et al. [83] obtain serum and plasma samples by centrifugation at 1200× g for 10 min and according to the results of this study, the concentrations of the miRNAs in plasma and serum were highly correlated. On the other hand, Wang et al. [78] followed a two-step centrifugation protocol to recover plasma and serum, i.e., the blood was initially centrifuged at 1000× g for 15 min and the resulted supernatant was centrifuged again at 2500× g for 15 min. The additional centrifugation step probably removed the platelet contaminants (that contain significant amounts of miRNAs) from plasma and therefore in this study higher concentrations of miRNAs were detected in serum compared to plasma. Finally, McDonald et al. [82] initially recovered plasma and serum by centrifuging samples at 795× g for 20 min and a higher concentration of miRNAs in plasma that in serum was observed. Subsequently, the serum and plasma samples underwent additional centrifugation steps, i.e., 15,000× g/10 min and 355,000× g/1 h. The miRNAs concentration was determined in the resulted supernatant of each centrifugation step. The additional centrifugation step at 15,000× g resulted in a significant decrease of miRNAs concertation in plasma but did not have an effect on miRNAs concertation in serum. Interestingly, further centrifugation at 355,000× g did not have an effect on miRNAs levels in both plasma and serum. Based on the results of this study it was suggested that cellular contaminants, especially platelets contribute miRNAs in plasma increasing their concentrations compared to serum.

Taken together, the results of the above examples suggest that the centrifugation conditions (including, g-force, time, temperature, etc.) that are applied to recover plasma or serum samples or to remove precipitates and other contaminants from samples prior to RNA extraction needs to be standardized as residual platelets, cell debris, etc., can alter miRNA levels [95]. Importantly, the separation of cell pellets from blood fractions must be ensured to avoid contamination with cellular miRNAs [85,98].

#### 3.1.4. Effect of Hemolysis

Several studies have demonstrated that hemolysis occurring during blood collection and processing has a substantial impact on the miRNA content in plasma/serum [85]. However, the majority of these studies focused on the effect of hemolysis on specific c-miRNAs, including miR-16, miR-24, miR-15b, and miR-451. For example, McDonald et al. [82] reported that the concentrations of miR-15b, miR-16, and miR-24 were significantly increased in hemolyzed samples due to their release from erythrocytes. Therefore, special attention should be given when miR-16 is used as an internal control (discussed further below). It should be pointed out that the levels of specific miRNAs, such as miR-16 and miR-451, vary depending on the degree of hemolysis because these miRNAs are abundant in RBCs [99]. Recently Blondal et al. [100] reported that the levels of many miRNAs (overall 119 miRNAs were studied) significantly varied in compromised serum/plasma compared to their levels in high-quality serum/plasma. Interestingly, the levels of several c-miRNAs that have been proposed as biomarkers for CVDs are affected by hemolysis. For example, Kirschner et al. [101] demonstrated that the levels of miR-92a, a potential miRNA biomarker for ischemic heart disease, in circulation are affected by hemolysis. On the contrary, the levels of the multifunctional miR-155, which is linked to CVDs including coronary artery disease, abdominal aortic aneurysm, HF, and diabetic heart disease [102] and miR-625 that is downregulated in hypertensive patients [32] are not affected by hemolysis. 

Hemolysis is usually estimated by measuring the absorbance of the plasma/serum samples at 414 nm which is the maximum absorption wavelength (λmax) of free hemoglobin [101]. However, this approach has many limitations in identifying hemolyzed samples. Thus, in the field of c-miRNA research, it has been proposed that specific miRNAs such as miR-451 could be used as the first indication of hemolysis. In addition, the extent of hemolysis can be evaluated by measuring specific miRNAs including miR-16, miR-15b, or miR-451 in order to decide whether a serum/plasma sample is suitable for the determination of the levels of miRNAs related to CVDs [101,103].

Together, the above results suggest that the levels of other potential miRNAs biomarkers could be affected by hemolysis. Even though in vivo hemolysis cannot be avoided, in vitro hemolysis could be significantly reduced by following the guidelines of blood collection and handling that have been proposed by the Early Detection Research Network (ERDN) [104]. Interestingly, it has also been demonstrated that the use of small diameter needles (23 gauge or above) should be avoided due to shearing induced hemolysis of RBCs which contain abundant miRNAs and can alter profiling results [105]. It has been recently proposed [96] that after evaluation of hemolysis, all samples have to be subjected to miRNA detection, but only non-hemolyzed samples have to be initially used for marker discovery. Subsequently, when miRNAs of interest have been determined, experimenters should examine whether the selected miRNAs are affected by hemolysis. If hemolysis does not have an impact on the levels of the selected miRNAs, then hemolyzed samples may also be included in the analysis.

#### 3.1.5. Sample Storage and Handling

Clinical studies aiming to identify c-miRNAs as biomarkers will probably compare retrospective archived serum/plasma with freshly collected samples. Subsequently, a standard operating procedure must be employed for the isolation, storage, and handling of such samples in uniform conditions [30]. It is well known that c-miRNAs can be detected in archival patient plasma and serum samples stored long-term at −80 °C as well as after treatment with RNases [106,107,108]. Interestingly, c-miRNAs in plasma are detected without significant changes in their expression levels after up to 8 freeze/thaw cycles [83]. However, c-miRNAs are less stable in serum compared to plasma [30,108,109] as repetitive freeze/thaw cycles affect both their levels and pattern. Importantly, the levels of each c-miRNA are affected in a different way and therefore the stability of each miRNA may be related to its cell-free form, i.e., whether it is membrane encapsulated, protein-bound, or unbound [30]. Thus, there is a critical need to develop a protocol for c-miRNA isolation from serum that reduces sample degradation without introducing factors inhibitory to downstream assays. 

#### 3.1.6. Effect of Thaw Temperature and Speed

Surprisingly, the thaw speed and temperature (i.e., rapid thawing at 37 °C vs. slow thawing at 4 °C) also affect the c-miRNAs level. Farina et al. [30] reported that two c-miRNAs, i.e., miR-93-5p and miR-451a, were detected at significantly higher levels when the serum was thawed rapidly at 37 °C compared to the miRNA levels that were detected when the samples were slowly thawed at 4 °C. It has been proposed that these miRNAs are contained within large particles that either sediment in the initial centrifugation step to remove cryoprecipitates when the serum is thawed at 4 °C or are lysed or disrupted at 37 °C, releasing c-miRNAs. These findings indicate that varying sample preparation and handling processes dramatically alters the profile of circulating miRNA in serum leading to discrepancies among different studies [30]. In accordance with the EDRN standard operation procedure for blood collection and handling [104], archived serum is stored at −80 °C, requiring at least one freeze/thaw cycle prior to c-miRNA detection.

#### 3.1.7. Repetitive Freeze-Thaw Cycles

Glinge et al. [92] reported that the levels of miR-1 and miR-2 decreased after four repetitive freeze-thaw cycles in both plasma and serum. A similar effect has also been reported by Zhao et al. [110] when they evaluated miR-346 and miR-134 levels; the levels of both miRNAs decreased in samples that were frozen/thawed. On the contrary, Farina et al. [30], demonstrated that the expression levels of several miRNAs including miR-21 increased after repetitive freeze-thaw cycles. These results suggest that a general guideline regarding the handling and processing of frozen samples is required because, as discussed above the thawing speed and temperature affect the miRNA profile in plasma/serum samples.

#### 3.1.8. Aliquoting Samples 

A major issue during the design of retrospective studies is how to carefully handle precious, and often irreplaceable, patient samples. Many archived samples are stored in volumes larger than required for a single RNA isolation. In addition, future detection of other biomolecules, such as DNA and protein, may be desired from the same sample. Thus, it is recommended that each serum/plasma sample be divided into 1 mL aliquots and unused aliquots be re-frozen to −80 °C at the time of RNA isolation. Proceeding with RNA isolation from at least 1 mL of serum followed by storage of RNA in small aliquots at −80 °C for future analysis eliminates multiple freeze/thaw cycles and precious patient samples are optimally utilized. Importantly for prospective studies, freshly isolated serum should be stored at −80 °C, at least overnight, to allow direct comparison to archival samples [30]. In addition, the storage time and temperature may also have an effect on the composition of miRNAs. McDonald et al. [82] reported that the level of several miRNAs, including miR-16, is changed after storage for 24 to 72 h at either 4 °C or −20 °C and leads to another issue in sample processing. Therefore, it is clear that different experimental setups, processes, and sample handling lead to the bias in the final results of miRNAs [28] and that validated and optimized experimental protocols are sorely needed. 

### 3.2. Effect of Extraction Method

miRNAs are relatively stable and can be reliably measured in tissues, as well as in biofluids [84]. However, isolation of miRNAs from plasma and serum is challenging, because of their low concentration and because contaminants from blood including heme and immunoglobulin G (IgG) may affect the results [88]. Moreover, as discussed above, miRNAs are transported in the plasma or serum via different mechanisms, such as within RNA-protein complexes or extracellular vesicles [83]. Different forms of stable miRNAs might possess different levels of resistance and vulnerability to particular isolation methods and previous studies on the exportation of miRNAs from cells raised the prospect that plasma and serum might exhibit some differences in their miRNA content [78]. 

A major issue of the studies of c-miRNAs is the low concentration of total RNA in bio-fluids, which makes it difficult to measure both the quality and concentration of isolated RNA using spectrophotometric methods (e.g., nanodrop) [111]. In CVD-patients, higher amounts of miRNAs can be released to circulation compared to healthy individuals. Thus, it has been proposed to use equal volumes of starting material (serum or plasma) instead of using the same amount of total RNA in order to have accurate results [79]. Moreover, variations in biofluids composition may introduce some variability across samples during miRNAs extraction and purification. To control these inter-sample differences, it has been suggested that a known amount of exogenous miRNA (e.g., from *Caenorhabditis elegans*) should be added at the beginning of the purification step (discussed further below) [83]. 

Another disadvantage of using miRNAs as biomarkers for clinical diagnosis is their laborious extraction and detection procedures while the methods that are currently used to isolate and estimate miRNAs levels require optimization [112]. In general, RNA extraction methods are divided into three major categories: (i) the phenol-based techniques that employ organic solvents, phase separation, while RNA is recovered by precipitation; (ii) the methods that initially use phenol/chloroform to isolate RNA from other biomolecules and subsequently a column for RNA adsorption; and (iii) the phenol-free methods that use a lysis buffer to release RNA in the solution and a column for RNA recovery [96,113]. Several commercially available kits that facilitate the process of RNA isolation have been widely used to recover miRNAs from blood samples. Despite several studies having tried to directly compare and optimize the extraction methods for miRNAs [82,96,114,115,116], several inter-laboratory differences in results still exist [117]. Several studies have investigated the effect of RNA isolation methods on miRNA recovery. Although there is no conclusion on which method is the best, there is an agreement that the different isolation methods provide different yields and/or quality of miRNA. In a recent study, Moret et al. [111] compared four different RNA extraction methods, i.e., TRIzol-LS, mirVana PARIS kit, miRNeasy Serum/Plasma kit, and a combination of TRIzol-LS and mirVana using both frozen and fresh plasma sample. The results of this study revealed that the organic/phenolic phase in the TRIzol-based methods had a negative impact on the quality of the extracted RNA and column-based methods should be preferred. Likewise, Sourvinou et al. [114] reported that higher RNA yields were obtained using two column-based RNA extraction kits (mirVana PARIS and miRNeasy Mini) compared to the TRIzol-based extraction method. 

Even though the advantages of column-based extraction methods over the TRIzol have been highlighted by recent publication, there is a discrepancy in the literature regarding the efficiency of the commercially available miRNA isolation kits (Table 4). For example, Kroh et al. [118] demonstrated that Qiagen miRNeasy kit provides 2-to 3-times higher RNA yield compared to Ambion mirVana. However, Kroh’s results were in contrast with those reported by Sourvinou et al. [114], showing that the mirVana PARIS kit provides higher RNA yields compared to the miRNeasy kit. It should be noted that in Khor’s work, the mirVana PARIS recommended—by the manufacturer— protocol was modified by introducing an additional organic extraction step, while the protocol of the miRNeasy kit was modified by using 10 volumes of Qiazol reagent (instead of 5 volumes that is recommended by the manufacturer) to initially denature 1 volume of plasma or serum. In another study, Li et al. [119] compared miRNA measurement results including RNA yield and amplification efficiency of plasma RNA using seven different commercially available kits as illustrated in Table 4. The results of this study revealed that despite synthetic RNAs were recovered by all kits (≥50% recovery), some kits (i.e., RNAdvance and miRCURY) displayed biases in the length of RNAs that could be isolated. On the contrary, MagMAX, Quick-RNA, DirectZol, miRNeasy, and mirVana kits showed comparable recovery regardless of the length of RNA. Moreover, the recovery of plasma miRNAs from some kits was extremely low whereas according to this study miRNeasy kit showed a better overall performance in terms of miRNA purity and recovery. Interestingly, miRNAs were not detected in samples obtained from the MagMAX kit while the Quick-RNA kit had a low amplification for miRNAs. Though, this finding needs to be confirmed in different laboratories with a larger and diverse number of samples and different classes of miRNAs. In a most recent study, Vigneron et al. [120] compared three commercially available isolation kits, i.e., miRNeasy Serum/Plasma, mirVana, and NucleoSpin miRNA Plasma, which had recently been made available. In this study, higher concentrations of miR-16-5p were obtained by the NucleoSpin kit compared to miRNeasy and mirVana kits. In addition, higher miR-16-5p concentrations were obtained by the miRNeasy kit compared to the mirVana kit from the same samples. In another study, Tan et al. [116] compared five commercially available miRNA extraction kits (Table 4) for the determination of spike-in miRNAs in plasma samples. The results revealed the concentration of spike-in miRNAs controls obtained from plasma samples using all the five kits were highly comparable. However, the lowest extraction bias across samples was observed in samples that were obtained using the NucleoSpin kit. On the contrary, McAlexander et al. [121] reported that the concentrations of miRNA spike-in controls obtained using the miRNeasy kit were slightly lower compared to those obtained using the miRCURY kit.

Taken together, the results of the aforementioned studies illustrate, that different miRNA extraction kits provide different quality of miRNAs in terms of purity, composition, and yield, making it difficult to directly compare the results of different studies that used different RNA isolation kits. It should be pointed out that the protocols are usually modified by each research group, resulting in the generation of inconsistent results between these studies. Therefore, these methods should be standardized and optimized by comparing different extraction techniques in the same study using different types and quantities of starting material [117]. Overall, further studies are needed to elucidate whether isolation kits differ in their overall effectiveness to isolate miRNA from blood samples and remove contaminants, as well as to examine whether each kit has preferences for specific RNA/miRNAs classes. It has been suggested that this type of test is essential to compare the results of different studies that used a different method or isolation kit to extract c-miRNAs as well as when limited information regarding the RNA isolation method is available [122]. Overall, the discrepancies among several studies are emphasizing the great need to optimize and standardize the isolation methods of miRNAs [113,123].

### 3.3. Detection Methods and Normalization Strategies

A variety of platforms have been developed to identify miRNAs and quantify the expression levels of c-miRNAs in plasma/serum samples including qRT-PCR, miRNA microarrays, or next-generation sequencing (NGS) [124,125]. Each method has its own advantages, disadvantages, and limitations and these techniques have been recently reviewed in [84]. The qRT-PCR-based methods have several advantages compared to other techniques, including cost-effectiveness, speed, but are limited by low throughputs, i.e., they can only analyze a small number of samples, as compared to microarrays. Microarray-based methods require higher amounts of starting material for analysis, and this may intercept the routine analysis of miRNA when low amounts of miRNA are obtained. Microarray-based methods are affected by the short length and similar sequence among families and clusters of mature miRNAs. In addition, microarrays require a pre-amplification step that may alter the real concentration of c-miRNAs [126]. On the other hand, NGS requires lower amounts of starting material (c-miRNAs) and because of its lower cost and higher throughputs is became a preferred method for c-miRNAs analysis. Importantly, NGS offers the advantage of identifying novel miRNAs. However, it is still difficult to quantify the levels of c-miRNAs due to their low concentrations while all the miRNA detection methods depend on the quality of the starting material. In addition, the most widely used methods including qRT-PCR and microarray are based on the assumption that the different RNA extraction methods isolate all miRNAs equally [29]. However, as discussed above the levels of c-miRNA isolated using different methods should not be directly compared as the recovery of c-miRNA varies dramatically [127,128]. Currently, qRT-PCR is the most widely used method for miRNA quantification. Therefore, the normalization of the qRT-PCR data is vital for the reliable quantification of the circulating miRNAs in the plasma and serum samples [103].

In general, two methods are usually employed to analyze data from qRT-PCR: (i) the absolute quantification approach that determines the input copy number, by comparing the PCR signal to a standard curve, and (ii) the relative quantification approach that relates the PCR signal of the transcript of interest to that of a control sample [129]. For the miRNA quantification using the latter method, the PCR-derived cycle threshold (Cq) of the target miRNA is compared with that of a stably expressed endogenous miRNA (and/or of an exogenous miRNA of known concentration) from the same sample. However, without added spike-ins and standard curves, all techniques are mostly based on relative quantification, and the levels of miRNAs are presented as a “fold change” between paired samples, and not as an absolute unit, introducing bias when results from different studies are compared. Normalization strategies usually lead to different results and therefore to avoid misleading interpretation of data it is essential to choose the most reliable normalization method in each experimental setting [130]. 

Circulating miRNA expression levels are usually normalized using endogenous and/or exogenous control to account for biological and/or isolation variabilities. Even though there is no well-established endogenous miRNA that could be used as an internal control, hsa-miR-16-5p, small nucleolar RNA U6, and total RNA concentration are frequently used in several studies [113]. However, miR-16-5p is highly expressed in RBCs and hemolysis contributes to a significant increase in its concentration in plasma and serum [82,101]. Therefore, special attention should be given if miR-16 is used as an internal control. In addition, U6 is less stable in serum/plasma samples from patients with various diseases including cancer [103] and CVDs [131]. A recent study also demonstrated that the combination of let-7d, let-7g and let-7i could act as a reference for the normalization of serum miRNAs and that is probably better compared to the reference genes miR-16 and U6, [132]. The endogenous reference genes that are routinely used for data normalization have been recently reviewed in [133]. However, until today a reliable “housekeeping” c-miRNA has not been identified [134], suggesting that correcting plasma/serum volume is the best method of normalization, as the volume of plasma/serum is clinically standard for other biomarkers [130]. Moreover, in a disease state including CVDs, higher amounts of miRNAs are released in the circulation compared to healthy individuals and thus equal volumes of starting material should be used instead of using the same amount of total RNA [79]. In addition, special attention is need when endogenous miRNAs are selected as an internal control because even though these miRNAs may be stable in some studies, they may change in other pathological conditions and are therefore not suitable as an internal control [130].

Another method that is widely used for normalization is the addition of synthetic spike-in miRNAs, mainly *Caenorhabditis elegans* miRNAs (cel-miR-39-3p, cel-miR-54-3p, and cel-miR-238-3p), without homology to human miRNAs at the beginning of RNA isolation [83]. However, it has been demonstrated that these exogenous miRNAs are unstable in crude plasma and thus the timing that the spike-in miRNAs are added to the plasma/serum samples is essential because the plasma/serum RNases should be fully inactivated [135]. It should be noted that the efficacy of exogenous vs endogenous normalizations has not been assessed yet. 

Based on the above data, normalization is probably one of the most important steps when determining c-miRNAs expression levels. Even though several reference genes have been proposed in analyzing circulating miRNAs further studies are needed to compare these different methods and identify the most reliable method of normalization, which might be specific for the release route of the miRNAs e.g., microparticles or protein-bound [27]. Moreover, it is essential to establish the optimal endogenous control for each type of CVD, because the expression profiles and/or expression levels of specific c-miRNA may change in different types of CVDs. To this end, several studies have proposed that for normalization of miRNA concentrations, multiple reference genes or a suitable combination, as well as a standard concentration of spike-in miRNAs, should be used. In addition, it has been proposed that all samples must be simultaneously processed using identical stating volumes (See [103] as well as reference cited therein).

### 3.4. Other Patient-Related Factors 

#### 3.4.1. Medication 

Previous studies have shown that the administration of heparin before blood collection affects the results of miRNAs [136] while endogenous heparin has also a significant impact on miRNA quantification [137]. Moreover, anti-platelet therapy also affects miRNAs profile and interestingly aspirin has an impact on the expression levels of platelet-derived miRNAs such as miR-19b and miR-92a [138]. Likewise, anti-platelet treatment also affects the expression levels of other platelet- miRNAs such as miR-191, miR-223, miR-126, and miR-150 [139]. Thus, special attention should be given during the recruitment of patients for studies of miRNA quantification with respect to heparin and anti-platelet administration before blood collection. It should be pointed out the treatment of blood samples with the enzyme heparinase reverses the effect of heparin [140]. Furthermore, medication such as angiotensin-converting enzyme (ACE) inhibitors or blockers might have an effect on the expression levels of specific miRNAs (miR-155, miR-19a, miR-378, miR-222, miR-342, miR-145, and miR-30e-5p) that are downregulated in CAD patients [64]. Thus, it is essential to examine the effect of drugs on the expression levels of miRNAs that have been implicated in the pathogenesis of CVDs. In addition, the inconsistency of the published data may be explained by studies that will examine the pathways of miRNAs release in blood circulation. For example, Gidlöf et al. [141] reported that cardiac miRNA levels are affected by renal function indicating that the renal function might also have an impact on the plasma levels of miRNAs.

#### 3.4.2. Physical Activity 

It is well known that physical activity (PA) affects the adaptive response that regulates the body’s homeostasis during exercise stimulation. The adaptive response is based on the alteration of gene expression that regulates a variety of physiological processes such as myocardial and skeletal muscle metabolism, regeneration and remodeling. Interestingly, several studies have shown that PA might prevent or delay the progression of several pathological conditions [142]. In addition, skeletal muscle changes its phenotype in response to various neuromuscular activity and several studies have investigated the effect of exercise, of different intensity and mode, on the expression profile of miRNAs in muscles [143]. The miRNAs most abundant in muscle tissue are miR-1, miR-133a, miR-133b, miR-206, miR-208a, miR-208b, miR-486, and miR-499, and these regulate of skeletal and cardiac muscle proliferation, differentiation, metabolism, and hypertrophy. It should be noted that miR-208a is cardio-specific, while miR-206 is skeletal muscle-specific [144]. Though future studies are needed to establish the role of miRNAs as mediators of the physiological processes including cardiac function that are affected by PA [143] and the physical status of individuals in miRNAs studies should be taken into account.

#### 3.4.3. Other Factors

Dietary constituents, e.g., vitamin A and D, curcumin, and many others affect the miRNA expression profile [145], while miRNAs contained in food may enter the circulation and alter the endogenous miRNAs concentrations [146]. In addition, co-variability of miRNA levels including miR-126-5p and miR-92a-3p with demographic factors such as age and serum creatinine level has been reported [147]. Another issue in miRNAs studies is that several miRNAs exhibit low tissue and disease specificity. For example, Witwer et al. [148] proposed that the increased levels of miRNA-141 which is increased in pregnant women, prostate cancer, and other cancers and CAD are released from epithelial, breast, colon and lung. Moreover, individuals’ physiological state and sampling biases such as the phlebotomy process can also contribute to variation in c-miRNAs levels among various studies (see [116] are references cited therein).

It has also been suggested that the blood collection time might have an impact on miRNA identification and quantification, because some components of the blood including HDL particles or circulating lipoproteins vary in fasting and non-fasting patients [103]. Moreover, since some of these blood components carry specific miRNAs, the blood collection time could contribute significantly to the variations in miRNA expression levels. Further studies to elucidate the effect of blood collection time on c-miRNAs concentrations are needed. 

## 4. Recommendations 

This review emphasizes the need to establish standard operating procedures including the choice of starting material, sample isolation protocol, detection, and normalization methods for the determination of miRNAs in circulation. Most of the factors that affect the expression levels and/or profile of miRNAs in circulation are summarized in Table 5.

Currently, there is no single definitive reliable approach in processing blood for c-miRNA assays; to that end, a detailed description of blood collection and processing methods in scientific publications must be reported [149]. Recently, different groups [92,150] have proposed a series of precautions/guidelines for handling blood samples including (i) the use of EDTA containing tubes (for plasma samples); (ii) the transfer of blood samples to the laboratory with minimal physical disturbance; (iii) the immediate centrifugation of blood samples; (iv) the very careful removal of the plasma afterward and immediate aliquoting of samples; (v) the storage of plasma/serum samples at −80 °C; and (vi) the avoidance of repetitive freeze-thaw cycles by using pre-aliquoted samples. Importantly, blood samples must be processed within a few hours of collection to restrict contaminating levels of miRNA expression derived from lysed RBCs, platelets, leukocytes while isolated plasma and serum samples should be carefully recovered and separated into aliquots and frozen within 24 h of separation. Furthermore, plasma/serum samples must be stored at −80 °C [92].

In addition, contamination of serum/plasma samples from cellular material as well as hemolysed samples should be avoided. It has been demonstrated that an additional centrifugation step, subsequent to plasma/serum recovery preparation, can effectively remove intact blood cells [103]. Importantly, mixing specimen types such as plasma and serum within a study must be avoided, because it is not clear whether miRNAs are expressed equally in these fractions. Overall, several pre-analytical and analytical variables must be considered in miRNA studies as follows: 

### 4.1. Sample Collection and Processing

Even though miRNAs in plasma/serum are stable at room temperature for up to 24 h [83], the time interval between blood collection and the subsequent processing of plasma or sera can cause a variation in the miRNA levels. As mentioned above, the cellular components of blood may also contribute miRNAs into the plasma/sera during the storage period and/or during the sampling and processing, and therefore it is recommended that samples should be processed within 2–6 h after collection [113].

### 4.2. Storage of Serum/Plasma Samples

Although storage of plasma or serum at −80 °C before analysis of miRNAs levels is recommended for comparison purposes with archival samples the effect of duration of storage at −80 °C on miRNA levels must be further examined. 

### 4.3. Extraction Method

The use of miRNAs as biomarkers for CVDs and other diseases is currently limited by the diversity of methodologies used for their extraction and detection. To reduce discrepancies among c-miRNAs studies a selective and consistent RNA isolation method should be employed.

### 4.4. Normalization

miRNA quantification is limited by the lack of standardized control transcripts for normalization. Even though several endogenous miRNAs are used as internal controls, their amount in a given plasma or serum sample may be affected by several factors including hemolysis, physical activity, fasting vs. non-fasting state at blood collection, etc. Despite this, only limited information is available regarding the effect of these factors on miRNAs expression levels in circulation, and therefore special attention should be given when miRNAs are selected as biomarkers or endogenous controls for research studies. Recently, the European Union-funded project (HOMAGE [Heart OMics in AGEing]) reported general guidelines when miRNAs in HF patients are measured and compared in the same sample sets of various laboratories to search for differences in technological approaches and data normalization (http://www.homage-hf.eu/).

## Figures and Tables

**Figure 1 ijms-21-00561-f001:**
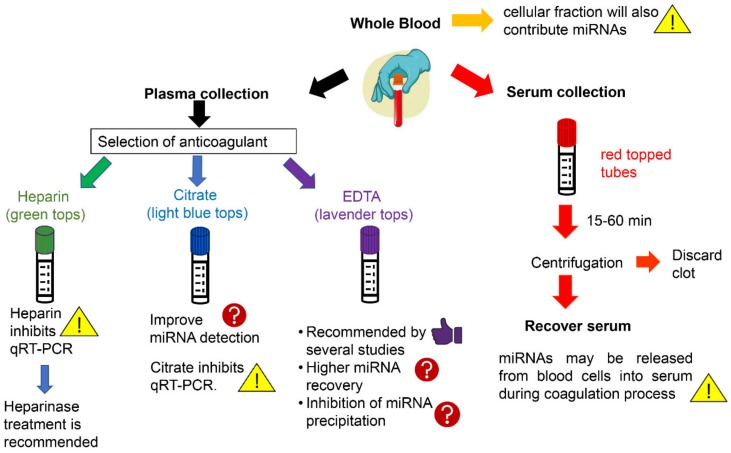
An overview of the effect of the different blood fractions and anticoagulants that are used for plasma isolation, on miRNA detection in circulation. The first step is the choice of blood fraction (whole plasma, serum, or plasma) to be used for miRNA detection. When plasma is selected then the effect of the anticoagulant on miRNA expression levels should be taken into account, and in general, heparin is not recommended for RNA studies, because of interference with downstream applications. Thumbs- up symbol (
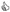
) indicates that EDTA should be used as an anticoagulant for the collection of plasma samples. Warning signs (!) show factors that have been verified experimentally by multiple studies to affect the detection of miRNAs in circulation, while question marks (?) illustrate factors that may have an impact on miRNAs detection and that should be examined further.

**Table 1 ijms-21-00561-t001:** Examples of different strategies for the determination of circulating miRNAs levels.

CVD	Source	Anticoagulant	Isolation Method	Controls	Detection Method	Ref.
CAD	Whole blood	N/A	PAXgene miRNA ^a^	RNU44	TaqMan q-RT PCR	[64]
CAD	Serum	N/A	QIAamp ^b^	None	Sybr-green q-RT PCR	[65]
MI	Plasma	EDTA	TRIzol LS ^c^	spike-in cel-miR-39	TaqMan q-RT PCR	[66]
MI	Plasma	Citrate	mirVana PARIS ^d^	RNU6	Sybr-green q-RT PCR	[67]
AMI	Serum	N/A	mirVana PARIS	spike-in cel-miRNAs	Sybr-green q-RT PCR	[68]
AMI	Plasma	EDTA	RNeasy Mini ^e^	spike-in cel-miR-39	TaqMan q-RT PCR	[69]
HF	Serum	N/A	NucleoSpin ^f^	miR-103-3p/miR-16	Sybr-green q-RT PCR	[70]
HF	Plasma	EDTA	TRIzol LS	spike-in cel-miR-39	TaqMan q-RT PCR	[71]

^a^ PAXgene miRNA (Qiagen, Hilden, Germany); ^b^ QIAamp (Qiagen, Hilden, Germany); ^c^ TRIzol LS (Thermo Fischer Scientific, Santa Clara, CA, USA); ^d^ mirVana PARIS (Thermo Fischer Scientific, Santa Clara, CA, USA); ^e^ RNeasy Mini (Qiagen, Hilden, Germany); ^f^ NucleoSpin miRNA plasma kit (Macherey-Nagel, Hoerdt, France).

**Table 2 ijms-21-00561-t002:** A comparison of miR-1 expression levels between AMI patients and healthy individuals from different research groups.

Cohort	Fraction	Extraction Method	Detection Method	[miR-1] Increase in AMI Patients (Fold *)	Ref
70 AMI/72 control	Plasma EDTA-treated	TRIzol LS (Invitrogen)	TaqMan qRT-PCR	~2.8	[73]
156 AMI/145 control	Plasma Citrate-treated	miRcute (Tiangen)	Sybr-green qRT-PCR	~60	[74]
117 AMI/100 control	Serum	Phenol/chloroform	TaqMan qRT-qPCR	~1.5	[75]

* Fold change of miR-1 expression levels in AMI patients compared to healthy individuals.

**Table 3 ijms-21-00561-t003:** Comparison of miRNAs expression levels in plasma and serum samples obtained from healthy individuals by three independent studies.

miRNA	Extraction Kit	Centrifugation (Force/Time)	Detection Method	Norm. Method	Outcome ^a^	Ref
miR-15b miR-16 miR-24 miR-451	miRNeasy (Qiagen)	1000× *g*/15 min & 2500× *g*/15 min	TaqMan qRT-PCR	Equal volumes of isolated RNA	Higher [miRNA] in serum	[78]
miR-15b miR-16 miR-24 miR-122	mirVana PARIS (Ambion)	795× *g*/20 min ^b^ 15,000× *g*/10 min 355,000× *g*/1 h	TaqMan qRT-PCR	Spike-in cel-miRs-39/54/238	Higher [miRNA] in plasma	[82]
miR-15b miR-16 miR-24 miR-19b	mirVana PARIS (Ambion)	1200× g/10 min	TaqMan qRT-PCR	Spike-in cel-miRs-39/54/238	[miRNA]_plasma_ = [miRNA]_serum_	[83]

^a^ In all three studies, plasma was isolated using EDTA as an anticoagulant ^b^ The effect of three different (sequential) centrifugation forces/times on miRNA profile in plasma and serum was studied.

**Table 4 ijms-21-00561-t004:** Comparison of commercially available miRNA extraction kits by various groups.

Extraction Kits	Sample	Controls	Outcome	Ref.
mirVana ^a^ miRNeasy ^b^ Norgen ^c^	Plasma	cel-miR-39 hsa-miRs-21/16	Highest miRNA yields obtained by mirVana	Sourvidou el al. [114]
miRCURY ^d^ miRNeasy NucleoSpin ^e^ mirVana Norgen	Plasma	cel-miRs-39/54	Comparable miRNA concentrations	Tan et al. [116]
mirVana miRNeasy	Plasma/serum	cel-miRs-39/54/238	~2 times higher RNA yield with miRNeasy	Kroh et al. [118]
RNAdvance ^f^ MAgMAX ^g^ miRCURY Quick-RNA ^h^ DirectZol ^i^ miRNeasy mirVana	Plasma	hsa-miRs-16/150	miRNeasy showed the best performance MAgMAX failed to amplify miRNAs	Li et al. [119]
NucleoSpin mirVana miRNeasy	Serum	miR-16-5p	Highest miRNA yield obtained by NucleoSpin	Vigneron et al. [120]
miRCURY miRNeasy	Plasma	has-miRs-16/150 cel-miR-39	Highest miRNA yield obtained by miRCURY	McAlexander et al. [121]

^a^ mirVana (Thermo Fisher Scientific, CA, USA); ^b^ miRNeasy plasma/serum (Qiagen, Hilden, Germany); ^c^ Norgen miRNA purification kit (Norgen Biotek Corp., Thorold, ON, Canada); ^d^ miRCURY-Biofluids (Exiqon, Vedbaek, Denmark); ^e^ NucleoSpin miRNA plasma kit (Macherey-Nagel, Hoerdt, France); ^f^ RNAdvance (Agencourt Bioscience, Beckman Coulter, Beverly, MA, USA); ^g^ MAgMAX (Thermo Fisher Scientific, CA, USA); ^h^ Quick-RNA (Zymo Research, Irvine, CA, USA); ^i^ DirectZol (Zymo Research, Irvine, CA, USA).

**Table 5 ijms-21-00561-t005:** Common factors affecting the evaluation of circulating miRNAs as biomarkers for cardiovascular diseases.

Sampling Factors	Analysis Factors	Biological Factors
Needle gauge, centrifugation conditions	Sample volume	Medication
Hemolysis	Extraction method (organic-based, column-based)	Physical activity
Storage/handling/freeze-thaw conditions	Extraction kit	Blood collection time
Blood fraction (serum vs plasma)	Detection platform (qRT-PCR, microarrays, NGS)	Dietary supplements
Anticoagulant (citrate, EDTA, heparin)	Normalization strategy (endogenous/exogenous controls)	Demographic factors (age, sex)

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
