# Peer review of "Challenges in Using Circulating Micro-RNAs as Biomarkers for Cardiovascular Diseases"

_ijms, 2020, doi:10.3390/ijms21020561_

Round 1

Reviewer 1 Report

The authors for sure qualify for this paper, which is a review.

I have two general comments:

As the authors correctly mention the paper ref 34, they need not to repeat what has been addressed in this review paper, or at least shorten their manuscript accordingly (following 2 paragraphs after the citation of ref 34).

While all their listings and comments on the current methodology are sound and correct, they cite a majority of papers that are not specifically related to CVD. I would expect these listings in close relevance to CVD rather to general knowledge on the current shortcoming in the methodology. These have been reviewed already in the past. This would make the review also more concise (and shorter, of course).

Thus a bit of rewriting seems required.

Author Response

IJMS-680108

Response to Reviewer 1 comments

The authors for sure qualify for this paper, which is a review.

We would like to thank the Reviewer for helpful and insightful comments. We made every effort to improve our manuscript accordingly and describe below all revisions made. All revisions have been highlighted in the revised paper to facilitate the review process. Point-by -point answers to your comments are given below.

I have two general comments:

As the authors correctly mention the paper ref 34, they need not to repeat what has been addressed in this review paper, or at least shorten their manuscript accordingly (following 2 paragraphs after the citation of ref 34).

We thank the Reviewer for this comment.  The two paragraphs following ref 34 (ref 40 in the revised manuscript) have been merged into one, while their length has been decreased according to the Reviewer’s comments (Lines 115-133).  

While all their listings and comments on the current methodology are sound and correct, they cite a majority of papers that are not specifically related to CVD. I would expect these listings in close relevance to CVD rather to general knowledge on the current shortcoming in the methodology. These have been reviewed already in the past. This would make the review also more concise (and shorter, of course). Thus a bit of rewriting seems required.

We thank the Reviewer for this comment. Indeed, in our review, we cite several papers that are not specifically related to CVDs. However, the main factors affecting c-miRNAs stability, determination, normalization, etc., in blood samples (including plasma and serum) have not been extensively studied in blood samples from CVD patients. On the contrary, the use of c-miRNAs as prognostic/diagnostic biomarkers has been extensively studied for other diseases, especially cancer. In addition, several papers examine the main factors affecting the miRNAs profile, stability, concetration, etc., using either blood samples from healthy individuals or spike-in (exogenous miRNAs). Therefore, we strongly believe that it would be beneficial to less familiar readers who might be seeing troubleshooting in using c-miRNAs for CVDs for the first time to have all issues related to the determination of miRNAs in circulation and specific guidelines to overcome these issues concentrated in one review. To the best of our knowledge, this will be the first review that summarizes the challenges in using c-miRNAs while at the same provides recommendations to overcome and/or prevent the main issues during the determination of miRNAs in plasma/serum samples from CVD patients. Furthermore, we strongly feel that the exclusion of the papers highlighting several issues during miRNAs analysis (but are not highly related to CVDs) will weaken our manuscript.

Reviewer 2 Report

In the review by Felekkis et al, the authors have discussed the challenges and hurdles in the use of circulating microRNAs as biomarkers for cardiovascular disease. They have provided detailed detection methods for circulating microRNAs in serum and plasma. Authors have discussed several parameters for study of circulating microRNAs in blood, plasma, and serum including sample preparation, methods of isolation and detection, effect of anticoagulants, effect of centrifugation, hemolysis, sample storage and handling, thawing temperature of the frozen samples and others. In addition, they have also discussed normalization methods for calculating the abundance of circulating microRNA by RT-qPCR methods including usage of different housekeeping genes and known microRNAs. Moreover, they have discussed how different parameters such as physical activity and medication can affect the levels of circulating microRNAs in serum and plasma of human beings. Finally, they have recommended different key points to study circulating microRNA which can serve as biomarkers in cardiovascular disease.

This review is nicely written and provides great details especially pros and cons of studying circulating micrRNAs in cardiovascular disease. However, I have some minor suggestions that can be incorporated to strengthen the review as follows:

I found a few redundant statements/comments that can be avoided.

On page 1 (line 37-39), it is not clear how miRNAs are secreted by high-density lipoprotein-HDL or nucleophosmin 1?

There are a few typos: In table 1, under detection method, it should be Taqman q-RT PCR not the “Tagman q-RT PCR”. On page 13 (line 499), it should be NGS requires……………not “NSG” requires.

Author Response

IJMS-680108

Response to Reviewer 2 comments

In the review by Felekkis et al, the authors have discussed the challenges and hurdles in the use of circulating microRNAs as biomarkers for cardiovascular disease. They have provided detailed detection methods for circulating microRNAs in serum and plasma. Authors have discussed several parameters for study of circulating microRNAs in blood, plasma, and serum including sample preparation, methods of isolation and detection, effect of anticoagulants, effect of centrifugation, hemolysis, sample storage and handling, thawing temperature of the frozen samples and others. In addition, they have also discussed normalization methods for calculating the abundance of circulating microRNA by RT-qPCR methods including usage of different housekeeping genes and known microRNAs. Moreover, they have discussed how different parameters such as physical activity and medication can affect the levels of circulating microRNAs in serum and plasma of human beings. Finally, they have recommended different key points to study circulating microRNA which can serve as biomarkers in cardiovascular disease.

This review is nicely written and provides great details especially pros and cons of studying circulating micrRNAs in cardiovascular disease. However, I have some minor suggestions that can be incorporated to strengthen the review as follows:

We would like to thank the Reviewer for helpful and insightful comments. We made every effort to improve our manuscript accordingly and describe below all revisions made. All revisions have been highlighted in the revised paper to facilitate the review process. Point-by -point answers to your comments are given below.

I found a few redundant statements/comments that can be avoided.

On page 1 (line 37-39), it is not clear how miRNAs are secreted by high-density lipoprotein-HDL or nucleophosmin 1?

We thank the Reviewer for this comment. The following paragraph and relevant references have been added in the revised manuscript (Lines 40-50). Further analysis regarding the secretion mechanism of miRNAs from HDL and/or NPM1 is beyond the scope of this review.

“The difference between protein-associated and vesicle-enclosed remains elusive. It has been proposed the secretion mechanism of each miRNAs depends on the cell type from which it is released as well its fate and function [10]. For example, miRNAs detected only in the HDL-associated fractions may be generated by cells with lipoprotein transport pathways. Interestingly, it has recently been demonstrated that inflammatory cells including macrophages, monocytes, and neutrophils export miRNAs to HDL as many of miRNAs that are highly abundant in these types of cells were detected in elevated concentrations in HDL [11]. miRNAs associated with other types of RNA-binding proteins including AGO2 and NPM1 may also be actively released from donor cells and taken up by recipient cells. On the contrary, miRNAs that are transported with vesicles may originate from specific cell types that produce high amounts of vesicles [12]. The biogenesis, function, and circulation of miRNAs have been extensively reviewed elsewhere [13-15].”

There are a few typos: In table 1, under detection method, it should be Taqman q-RT PCR not the “Tagman q-RT PCR”. On page 13 (line 499), it should be NGS requires……………not “NSG” requires.

The above typos have been corrected in the Revised Manuscript